# Nanostructured Silicon Enabled HR-MS for the Label-Free Detection of Biomarkers in Colorectal Cancer Plasma Small Extracellular Vesicles

Sanduru Thamarai Krishnan [1,2], David Rudd [1,2,*], Rana Rahmani [1,3], E. Eduardo Antunez [1,2,4], Rajpreet Singh Minhas [1,2], Chandra Kirana [5,6], Guy J. Maddern [5,6], Kevin Fenix [5,6], Ehud Hauben [5,6] and Nicolas H. Voelcker [1,2,7,*]

1. Drug Delivery Disposition and Dynamics, Monash Institute of Pharmaceutical Sciences, Monash University, Parkville 3052, Australia
2. Melbourne Centre for Nanofabrication, Victorian Node of the Australian National Fabrication Facility, 151 Wellington Road, Clayton 3168, Australia
3. Department of Microbiology, School of Biology, College of Science, University of Tehran, Tehran 1417935840, Iran
4. Centre for Research in Engineering and Applied Sciences, The Autonomous University of the State of Morelos, Av. Universidad 1001, Col. Chamilpa, Cuernavaca 62209, Morelos, Mexico
5. The Basil Hetzel Institute for Translational Health Research, Woodville 5011, Australia
6. Discipline of Surgery, Adelaide Medical School, Faculty of Health and Medical Sciences, The University of Adelaide, Adelaide 5000, Australia
7. Commonwealth Scientific and Industrial Research Organization (CSIRO), Clayton 3168, Australia
* Correspondence: david.rudd@monash.edu (D.R.); nicolas.voelcker@monash.edu (N.H.V.); Tel.: +61-399050308 (N.H.V.)

**Abstract:** Despite improvements in treatment options for advanced colorectal cancer (CRC), survival outcomes are still best for patients with non-metastasised disease. Diagnostic tools to identify blood-based biomarkers and assist in CRC subtype classification could afford a means to track CRC progression and treatment response. Cancer cell-derived small extracellular vesicles (EVs) circulating in blood carry an elevated cargo of lipids and proteins that could be used as a signature of tumour suppressor/promoting events or stages leading up to and including metastasis. Here, we used pre-characterised biobanked plasma samples from surgical units, typically with a low volume (~100 μL), to generate and discover signatures of CRC-derived EVs. We employed nanostructured porous silicon (pSi) surface assisted-laser desorption/ionisation (SALDI) coupled with high-resolution mass spectrometry (HR-MS), to allow sensitive detection of low abundant analytes in plasma EVs. When applied to CRC samples, SALDI-HR-MS enabled the detection of the peptide mass fingerprint of cancer suppressor proteins, including serine/threonine phosphatases and activating-transcription factor 3. SALDI-HR-MS also allowed the detection of a spectrum of glycerophospholipids and sphingolipid signatures in metastatic CRC. We observed that lithium chloride enhanced detection sensitivity to elucidate the structure of low abundant lipids in plasma EVs. pSi SALDI can be used as an effective system for label-free and high throughput analysis of low-volume patient samples, allowing rapid and sensitive analysis for CRC classification.

**Keywords:** high-resolution mass spectrometer; small extracellular vesicles; exosome detection; colorectal cancer diagnosis; label-free omics and surface assisted-laser desorption/ionisation

## 1. Introduction

Blood-circulating small extracellular vesicles (EVs) offer a potential biochemical signature in exploring the cross-talk between cancerous and healthy cells. EVs are nano-sized vesicles ranging between 30 and 150 nm, which contain bioactive compounds such as nucleic acids (DNA, microRNA and mRNA), proteins (including chemokines), cytokines

and growth factors enclosed in a lipid bilayer. Compared to healthy proliferating cells, colorectal cancer (CRC) cells often release excess EVs, carrying tumour derived molecules [1]. EVs are involved in angiogenesis (formation of new blood vessels) in cancer progression by transporting various pro-angiogenic molecules such as matrix metalloproteinases, vascular endothelial growth factors and microRNAs [2]. Additionally, exosomal lipid rafts are bound to integral proteins, surrounded by membrane lipids, and these lipids are major contributors to protein sorting, cell–cell communication and cancer cell recognition in CRC progression [3].

The nature of EVs found in the blood is directly related to molecular changes in CRC cells, rendering EVs an effective probe for longitudinal clinical follow-up in metastatic CRC prognosis [4]. Evidence has shown that several tumour relevant proteins can be detected on the surface of the circulating EVs that may facilitate blood-based rapid CRC diagnosis (liquid biopsy) [5,6]. Specifically, EVs-receptor proteins are of importance to differentiate stages of malignant tumours and progression towards liver metastasis. Belove et al. identified a subset of human leukocyte antigen complexes and 10 receptor proteins, including CD5, CD31, CD44, which were differentially expressed in EVs derived from CRC cell lines (LIM1215 and MEC1); importantly, these proteins were not detected in healthy serum samples [7]. Studies using mRNA and protein expression profiling methods have illustrated that CRC cell-derived EVs contain many mRNAs or proteins associated with oncogenes or metastasis-related genes [8–10], suggesting that EVs might be connected to the liver metastasis of CRC. However, no direct evidence for EVs involved in CRC metastasis has previously been provided. Therefore, this study is designed to discover possible signatures in the promotion CRC to liver metastasis.

CRC-derived EVs found in biological fluids are enriched in lipids such as cholesterol (CL) (55%), sphingomyelin (SM) (28%), hexosyl-ceramide (5.5%), phosphatidylserine (PS) (7%), phosphatidylcholine (PC) (2%), phosphatidylinositol (PI) (1%), and phosphatidylethanolamine (PE) (0.6%) [11]. A mass spectrometry-based lipidomics analysis conducted by Lydic et al. identified more than 520 lipid species upregulated in EVs secreted by the CRC cell line LIM1215. Among them, plasmalogen, ceramide, sphingomyelin subspecies and glycerophospholipids, such as phosphatidylcholine and phosphatidylserine, were enriched in LIM1215 derived EVs [12]. Currently, most CRC-relevant proteins and lipid-based biomarkers in blood-circulating EVs are yet to be comprehensively characterised. A major challenge in detecting CRC signatures in blood is the dominance of major proteins, such as albumin and platelet factors, which necessitates selective fractionation [13]. However, fraction collecting methodologies that are mass spectrometry compatible are not readily reproducible, in regard to major protein depletion, or sufficiently selective [14]. In addition, once purified, no definitive biological methods are specific enough to differentiate exosomal biomarkers for metastatic CRC diagnosis from normal endogenous EVs [15]. Fit-for-purpose diagnostic techniques for sensitive and selective analysis of CRC derived EVs are urgently required.

Advances in mass spectrometry and iterative improvements in multi-omics analysis have enabled comprehensive molecular analysis of multiple biomarkers to detect early signs of disease. Matrix-assisted laser desorption ionisation is a widely used analytical approach for sensitive detection (down to femtomolar) of biomolecules such as lipids, proteins or peptides with the aid of a co-crystallising chemical matrix (e.g., $\alpha$-cyano-4-hydroxycinnamic acid). MALDI-MS offers excellent sensitivity and can be applied rapidly to 'phenotype' low sample volumes. However, when looking at the combination of lipids and peptides/proteins, the addition of the co-crystallising matrix can mask the detection of target groups (lipids) and short sequence peptides from tryptic digestion. To circumvent this issue, nanostructured surfaces have been created that take on the role of the matrix, absorbing and diffusing laser energy for photoionisation.

Surface-assisted laser desorption/ionisation (SALDI) is emerging as a promising tool for protein and small-molecule analysis [16]. pSi SALDI substrates such as nanostructure-initiator mass spectrometry (NIMS) and desorption/ionisation on silicon (DIOS) comprise

distinct surface chemistry, pore size and depth, enabling accurate identification of the spectrum of molecules that differ in size. Fourier-transform ion cyclotron resonance-high resolution-mass spectrometry (HRMS) enabled SALDI substrates offer high mass spectral performance for molecular analysis [16,17]. Additionally, the performance of SALDI can be enhanced by implementing lithium adduct materials added to plasma samples. Lithiated-lipids reduce spectral complexity, specifically when detecting low abundant lipids in samples with limited sample volume.

This study used SALDI-HR-MS to detect CRC-associated proteins (using a bottom-up analysis), and lithiated lipid subsets in EVs, isolated from ~100 μL banked CRC plasma. Two pSi SALDI substrates: (a) NIMS using pSi substrates with an average pore size of 100 nm (in diameter) and 3.5 μm in-depth (i.e., thickness), and (b) DIOS using pSi substrates with a pore size of 120 nm and 450 nm in-depth, were employed to detect proteins and lipids, respectively, in a low volume of patient sample. The schematic representation shows the workflow of the method used to identify disease-relevant signatures in plasma EVs (Supplementary Materials, Figure S1). SALDI-HR-MS identified peptide ions to match against known tumour suppressor or promoting proteins, such as serine/threonine phosphatases (PP2A) and activating transcription factor-3 (ATF3), implicated in metastatic CRC. Lithium-consolidated analysis revealed that a range of exosomal lipid markers, including phosphatidylcholine, ceramide and phosphatidylinositol subsets, may be combined with EVs protein markers to derive a staged signature of CRC. In conclusion, SALDI-HR-MS represents a promising analytical tool for rapidly diagnosing metastatic CRC using low volumes (~100 μL) of banked plasma.

## 2. Materials and Methods

Blood plasma samples were received from The Queen Elizabeth Hospital (TQEH), South Australia, with the approval of the TQEH Human Research Ethics Committee (HREC) and the Monash University HREC. We have categorised sample groups into (i) cancer free individual (CFI) and healthy participants, (ii) CRC—contain CRC samples stage I to IV (stage IV CRC metastasis to any organ except liver) and (iii) CLM—colorectal cancer liver metastasis. A total of 27 samples were used for the analysis, including 11 CFI samples, 10 CRC samples and 6 CLM samples. All samples were de-identified and received in a closed container maintained at $-80\,^{\circ}$C. The plasma samples were thawed to $20\,^{\circ}$C before performing EVs isolation.

### 2.1. Isolation of Circulating EVs from Plasma Samples

According to the manufacturer's recommendation, the plasma EVs were isolated using the total exosome isolation kit (Invitrogen 4478359). The EVs were isolated from 100 μL of blood plasma by adding 50 μL of exosome isolation reagents and vortexed for 2 min. The mixture was incubated at $4\,^{\circ}$C overnight. Samples were then centrifuged at $10,000\times g$ for 1 h at $4\,^{\circ}$C. After centrifugation, the supernatant was discarded, and the pellets containing EVs were resuspended in 200 μL aliquots of phosphate-buffered saline (PBS) for protein and lipid analysis and stored at $-20\,^{\circ}$C.

### 2.2. Transmission Electron Microscopy Phosphotungstic Acid (TEM PTA) Negative Staining Protocol

The EVs morphology was characterised by TEM PTA negative staining. Initially, the total EVs protein concentration was measured using a Pierce™ Rapid Gold BCA Protein Assay Kit. Bovine serum albumin (20 μL solution, concentration from 20 to 2000 μg/mL) was used for calibration—the total protein concentration by measuring absorbance at 480 nm wavelength. The CRC proteins were diluted to 20 μg/mL in PBS for TEM analysis. Copper grids of 400-mesh coated with carbon film (EMSFCFTH400H-CU ProSciTech, Qld, Australia) were glow discharge treated in nitrogen using a Pelco easiGlow (Ted Pella Inc., Redding, CA, USA) to render the grids hydrophilic. A 3 μL aliquot of the sample was pipetted onto each grid, allowed to remain for 10 min and the excess fluid blotted away, the

sample was fixed with 3% glutaraldehyde in Sorenson's phosphate buffer, pH 7.2 (# 16539-45 ProSciTech, Qld, Australia) for 2 min, the excess blotted and replaced with 2 washes of Sorenson's buffer (# ASOR68C ProSciTech, Kirwan, QLD, Australia) at working dilution, blotted, then stained with 2% aqueous phosphotungstic acid pH 6.9 for 1 min, excess fluid was blotted and air-dried. The samples were examined using a Tecnai 12 Transmission Electron Microscope (FEI, Eindhoven, The Netherlands) at an operating voltage of 120 kV. Images were recorded using an FEI Eagle 4k × 4k CCD camera and AnalySIS v3.2 camera control software (Olympus, New South Wales, Australia).

## 2.3. Nanoparticle Tracking (NTA) for EVs Size Characterisation

The size distribution of isolated EVs was characterised using NanoSight NS300 (Malvern Panalytical, New South Wales, Australia. A 10 μL aliquot of EVs sample was mixed with 990 μL filtered PBS solution (0.22 μm filter) and diluted 1:1000 ratio for improved characterisation. The analysis was performed 5 times (60 s each) with 25 frames per s. For NTA analysis, the detection threshold was set at 2, camera level 8, and a media dilution of 1:1000, and the same settings were used throughout all experiments.

## 2.4. Extraction of Lipids and Proteins from EVs

The Folch method was used to extract exosomal lipids. Briefly, 100 μL of each aliquoted sample was lysed using a probe sonicator for 3 × 10 s. Two mL of a mixture of methanol: chloroform (1:2 *v/v*) was added directly to the 100 μL of the lysed samples. The homogenous mixture was vortexed and incubated for 20 min. Following mixing, a 0.9% NaCl solution was added to each sample. The samples were then centrifuged for $500\times g$ for 15 min. After phase separation, the lower phase containing lipids in chloroform was transferred to a new amber tube and allowed to dry under $N_2$ gas. The dried lipid extracts were resuspended in chloroform (5 mg/mL of final concentration) and stored at −80 °C for further analysis.

Albumin is the most abundant protein in blood plasma, which often interferes with untargeted protein analysis, and it is hence crucial to perform an albumin reduction step [18]. Briefly, 100 μL of each EVs sample was lysed using a probe sonicator, 3 × 10 s, to extract the proteins encapsulated by the exosomal lipid bilayers. One hundred μL of the lysed samples were incubated with 1% (by weight) of trichloroacetic acid (Merck Pty. Ltd., Bayswater, Australia) in isopropanol at a ratio of 1:10 (sample to organic solvent), which has been measured to remove ~95% of human serum albumin effectively. After initial albumin removal, the samples were vortexed and centrifuged at $1500\times g$ at 4 °C for 5 min. Following centrifugation, the supernatant was discarded, and the pellets were washed with 200 μL of HPLC grade methanol (Sigma-Aldrich, St. Louis, MO, USA). Following this, the samples were vortexed and centrifuged at $600\times g$ at 4 °C for 3 min. The supernatant was removed, and the pellet was dried completely by incubation at 95 °C for 30 s. The pellets were resuspended in 100 μL of 100 mM Tris-HCl, pH 8.1, and incubated at 95 °C for 10 min to denature the remaining intact proteins. Alkylating and reducing agents (10 mM iodoacetamide and 40 mM tris(2-carboxyethyl)phosphine hydrochloride), prepared in Tris-HCl, pH 8.1 buffer, were added for 10 min at 95 °C in order to reduce disulfide bonds and alkylate the resulting thiols. Before performing tryptic digestion, the total protein concentration was measured using the Pierce™ Rapid Gold BCA Protein Assay Kit (ThermoFisher, New South Wales, Australia). Following protein quantification, the tryptic digestion was accomplished by adding sequence grade modified trypsin (Promega-#V5111) to the proteins, for 20 μg of protein 1 μg of trypsin was added to each sample. The protein samples were digested overnight (16 h) at 37 °C (in a heat block) while shaking at 500 *g*. The tryptic digestion was stopped by adding 20 μL of 5% formic acid in water (vol/vol) to the samples. The desalting procedure was achieved using C18 stage tips (inhouse Whatman® C18 Paper purchased from Sigma Aldrich). The column was activated using 40 μL of methanol, centrifuged at $1200\times g$ for 2 min and washed with 40 μL of 0.1% formic acid in water. After activating the C18 column, the peptide samples were loaded into the stage tip,

centrifuged at $1200\times g$ for 5 min and washed three times with 50 μL of 0.1% formic acid by centrifuging at $1200\times g$ for 2 min. The peptide elution step was achieved by adding 50 μL of 70% acetonitrile and 30% of 0.1% formic acid in the water, centrifuged at $800\times g$ for 5 min, and collected in clean Eppendorf LoBind tubes for further analysis.

### 2.5. DIOS Fabrication

DIOS surfaces were fabricated using phosphorous doped (n-type) monocrystalline (500–550 μm thickness, 0.008–0.02 Ωcm) silicon wafers purchased from Siltronix Silicon Technologies, Archamps, France. The electrochemical etching procedure was carried out by the previously described method [19]. A light-assisted anodic etching was achieved in ethanol: hydrofluoric acid (HF) 1:1 ratio (25% $v/v$) electrolyte solution. A Teflon cell (2.25 cm diameter) was used to etch the silicon wafer, assembled with the cathode (gold) and anode (platinum) materials. The electrical circuit was activated by generating a density current of 2.24 mA/cm$^2$ for 3 min using Keithley 2460 SourceMeter (Tektronix Inc., Beaverton, OR, USA). The freshly etched pSi SALDI surface was washed with ethanol and dried under $N_2$ gas. The pSi surface was oxidised with ozone (0.5 L/min $O_2$) at a flow rate of 5 g/m$^3$ for 3 min using UV-1 UV-ozone cleaner (SAMCO Inc., Kyoto, Japan). Following oxidation, a pore broadening etch was performed by immersing the porous Si surface in 5% ($v/v$) HF/$H_2O$ aqueous solution for 1 min. Subsequently, a second ozone oxidation was performed. The oxidised pSi surfaces were functionalised by adding 90 μL of neat fluorinated silane (1H,1H,2H,2H-perfluorooctyltriethoxysilane) at 80 °C for 60 min. After functionalisation, the porous Si surfaces were washed using acetone and dried under $N_2$ gas. Following fabrication, a 1 μL of sample was spotted for each sample in triplicate to test performance. Each sample contain approximately 0.3 to 0.4 μg/μL of lipids (based on internal standard lipids)

### 2.6. NIMS Fabrication

A wet electrochemical etching method was used for NIMS fabrication [20]. A monocrystalline (500–550 μm thickness, 0.008–0.02 Ωcm) phosphorus-doped n-type silicon wafers (Siltronix Silicon Technologies, France) was used. Etching employed an electrolyte solution containing 25:200:1 volume ratio of HF/distilled water/surfactant (NCW1001 Wako Pure Chemical Industries, Osaka, Japan). Using a Keithley 2460 current source meter (Tektronix, Inc., OR), a constant current density of 70 mA/cm$^2$ was applied across the etching cell (area of 7 cm$^2$) for 200 s. The resulting pSi surfaces were then washed rigorously with water followed by ethanol rinsing and dried under $N_2$ gas. The freshly etched surfaces were ozone-oxidised at 0.5 L/min oxygen at a flow rate of 5 g/m$^3$ for 30 min using UV-1 UV-ozone cleaner (SAMCO Inc., Kyoto, Japan). The hydroxyl-terminated pSi surfaces were functionalised by adding 90 μL of neat bis-heptadecafluoro-1,1,2,2-tetrahydrodecyl)tetramethyldisiloxane (Bis17) at room temperature for 60 min. Following fabrication, a 1 μL of sample was spotted for each sample in triplicate to test surface performance. Each sample contain approximately 0.2 to 0.3 μg/μL of protein (based on BCA assay).

### 2.7. FT-ICR-MS Measurement and Sample Deposition

EVs lipid and peptide spectra were acquired using a Bruker Solarix XR 7T MALDI FT-ICR-MS (Bruker Daltonics, Billerica, MA, USA) equipped with SmartBeam II ultraviolet laser operated in positive ion mode. Data acquisition was performed through imaging analysis on the sample spots to eliminate 'hot-spots' in surface deposition, with a mass range of 200 *m/z* to 1200 *m/z* for lipids and from 500 *m/z* to 4000 *m/z* for peptides. External calibration using red phosphorus clusters was run prior to each analysis for mass accuracy, assigned as $\pm$ 1.5 ppm for peptide analysis and $\pm$ 0.6 ppm for lipid analysis. The analyte spots were imaged for 100 μm spatial resolution and analysed in SCiLS Lab software 2019b version (Bruker Daltonics, Germany).

For peptide analysis, 1:1 (*v/v*) peptide sample/TA30 solvent (30:70 *v/v* of acetonitrile: 0.1% trifluoroacetic acid in $H_2O$) was mixed, and 0.5 μL of the sample mixture was spotted onto the NIMS surface.

Lithiated cationisation $[M + Li]^+$ is a facile method to resolve interference of naturally abundant ions such as $Na^+$ and $K^+$, and interpretation of the long-chain base and the fatty acyl substituents of lipids [21]. Initially, a 0.5 mM lithium chloride (LiCl) aqueous solution (*w/v*) was prepared in HPLC grade methanol and mixed with the 1:1 (*v/v*) lipid sample. Subsequently, 0.5 μL of the sample mixture was spotted onto a DIOS surface.

## *2.8. Data Processing and Statistics*

In this study, we have categorised CRC patients into two groups (i) CRC and (ii) CLM. The data processing was performed in SCiLS lab software (Bruker Daltonics). Mass spectra were processed for baseline subtraction using the iterative convolution algorithm and normalised based on their own total ion count (TIC). The classification model was generated using the linear discriminant analysis algorithm model supported by SCiLS Lab software. Subsequently, a discriminative analysis was performed between (a) CRC vs. CFI and (b) CLM vs. CFI. The receiver operating characteristic (ROC) estimates specificity and sensitivity for threshold classifiers to quantify the diagnostic ability of charged ions that discriminates between sample groups. The top abundant *m/z* discriminates groups were selected based on the spectral values evaluated from 0.7 to 1 using the area under the ROC curve (AUC value) [22]. The top mass spectral features were selected and exported in CSV format for molecular interpretation (Figure 1, Panel 1).

SCiLS lab software has an inbuilt advanced machine learning algorithm for data pre-processing and filtering—these are peak binning, peak picking, baseline subtraction and post-processing statistical methods. Peptide quantitation was not performed for peptide ions as this type of analysis can be consider to be similar to label-free quantitation. However, before performing trypsin digestion, BCA estimation of protein concentration was performed—showing each sample had 200 μg/mL ± 30 S.D. of protein. The peptide spectral values were incorporated into the MASCOT database for protein identification. The search query for peptide signals were performed by selecting parameters such as Swiss-Prot for database and Homo sapiens within the taxonomy function. In the bottom-up tryptic digestion, carbamidomethylation was selected as a fixed modification (post-translational modification introduced to the cysteine residue), and the variable modification was selected as acetyl protein N-terminal and oxidation (M) residue. The mass accuracy was confirmed based on the monoisotope of each peptide, assigned with ±2.5 ppm mass tolerance and designated as the singly charged ion, $[M + H]^+$, or conventional adduct $[M + Na]^+$, $[M + K]^+$.

The Lipidmaps database was used to identify corresponding lipids matched to the top abundant *m/z* values identified from the discriminative analysis. The search query was executed with a mass tolerance of ±0.005 Da of each *m/z* value for lithiated lipid ions $[M + Li]^+$.

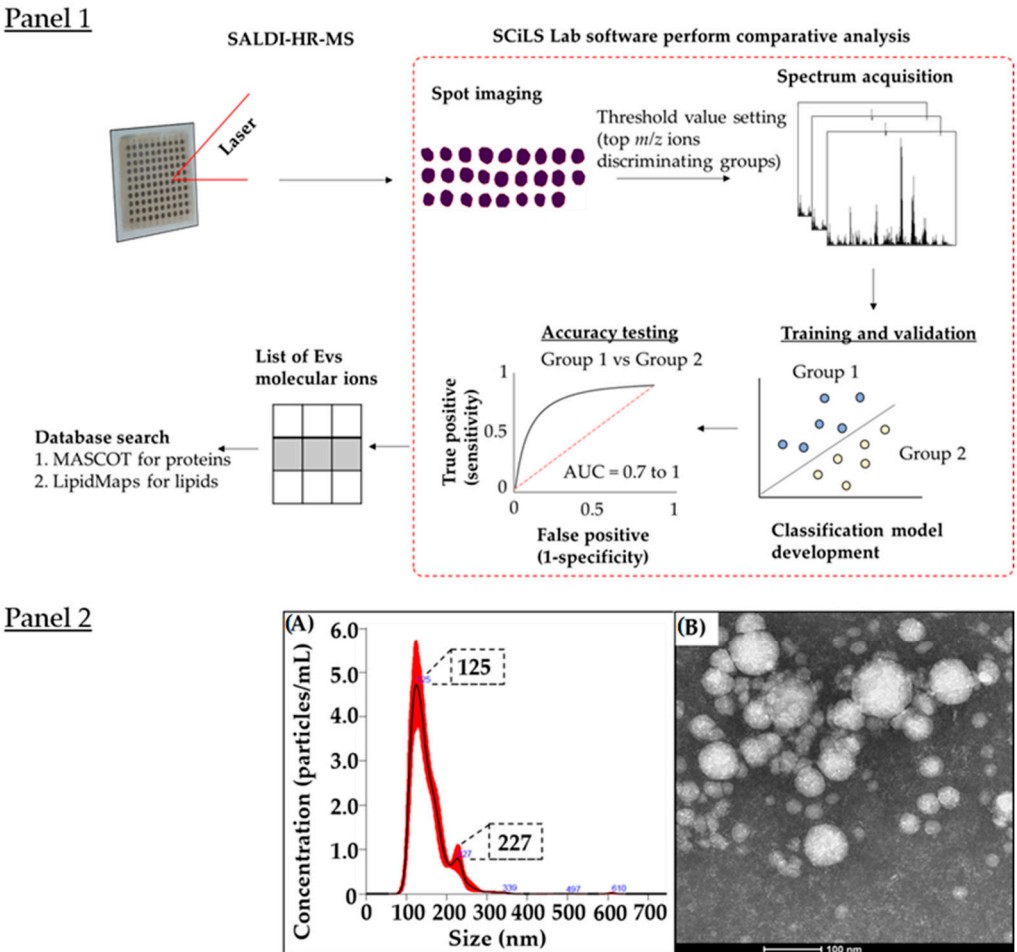

**Figure 1.** (**Panel 1**) illustrates the workflow and statistical analysis to discover the top discriminative spectral ions from CRC, CLM and CFI groups. Firstly, each sample's spots were selected, then threshold values for the top 50 $m/z$ spectral ions were assigned. Secondly, the linear discriminant analysis algorithm model classifies different sample groups (for example, CRC vs. CFI). The algorithm selected $m/z$ spectral values ranging between AUC 0.7 to 1. Lastly, a list of top spectral ions peaks that discriminate sample groups were acquired. These ions were then uploaded into databases including MASCOT and LipidMaps to identify putative proteins and lipids. (**Panel 2**) shows size of EVs isolated from CRC plasma samples. (**A**) NTA method identified that the EVs concentration was $3.27 \times 10^8 \pm 1.53 \times 10^7$ particles/mL with the size of Mean: $152.0 \pm 1.1$ nm, Mode: $125.7 \pm 5.8$ nm, SD: $43.1 \pm 2.1$ nm in size. (**B**) Negative staining TEM that plasma-derived EVs ranged between 40 to 120 nm size with spheroid shape. CRC—Colorectal cancer stage-I to stage-IV; CLM—Colorectal cancer liver metastasis; CFI—Cancer free individual; NTA—Nanoparticle tracking analysis; TEM— Transmission electron microscopy; EVs—Small extracellular vesicles.

## 3. Results

### 3.1. EVs Characterisation

The EVs size and concentration were determined using NTA, which captured $14.7 \pm 0.6$ of particles/frame at the concentration of $3.27 \times 10^8 \pm 1.53 \times 10^7$ particles/mL, and confirmed that the isolated EVs were on average $125 \pm 1$ nm in size (Figure 1, Panel 2). TEM characterisation detected EVs sizes ranging between 40 nm and 120 nm, mostly with a spheroid shape. The EVs size detected by NTA is larger (~20% increase in size relative to the average measured using TEM) potentially due to low levels of aggregation between EVs and smaller extracellular vesicles (20–45 nm diameter), which would be retained in the isolation procedure. Sample spheroid-shapes were checked prior to use (Figure 1,

Panel 2). Subsequently, the isolated EVs were subjected to SALDI enabled HR-MS for the determination of disease discriminative fingerprints.

### 3.2. SALDI Surfaces for Lipid and Peptide Detection

Electrochemical anodisation was performed to generate NIMS and DIOS substrates. The average pore size of NIMS substrates was 100 nm the thickness of the porous layer 3.5 µm (Figure 2A,B). The average pore size of DIOS was approximately 120 nm in diameter and 450 nm in-depth (Figure 2C,D). All analyte spots were dried under a gentle stream of nitrogen before performing MS analysis. The advantage of using NIMS and DIOS is the ease of fabrication and the ability to detect low abundant molecules [23]. Based on previous studies, NIMS performs well to detect peptides in biological samples [19,23–25]. NIMS uses trap liquid initiator material-Bis17, released upon heating by laser irradiation, which carries absorbed peptide molecules. Previously, NIMS has been used to detect peptide hormones, specifically growth hormone-releasing peptide-6 (GHRP-6) and growth hormone-releasing peptide-2 (GHRP-2), in spiked urine substitute (Surine™) [19], at physiologically relevant concentrations. Several studies have used DIOS for the detection of small molecules, such as lipids and metabolites, with ranges <700 Da [23,26]. For instance, Guinan et al. performed metabolite (oxycodone, *m/z* 316) detection in human saliva and blood using DIOS. The signal intensity ratio obtained by DIOS substrates was larger compared to other substrates such as NIMS and showed excellent sensitivity with detection limits in the low and sub ng/mL range [27].

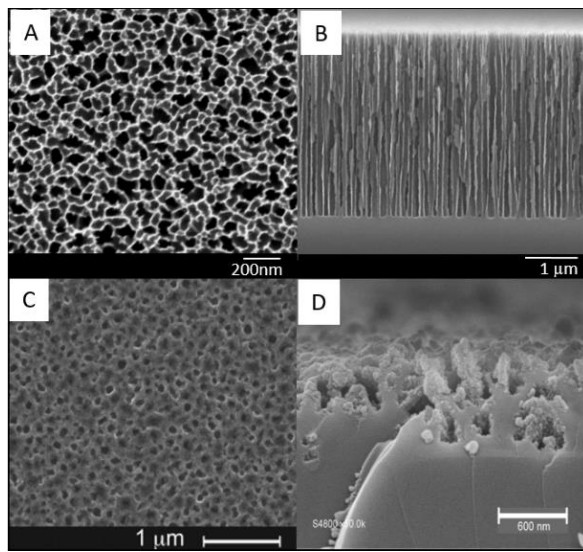

**Figure 2.** The NIMS surface morphology is shown by scanning electron microscopy in the top and cross-sectional views (**A,B**). The pore size of NIMS was estimated between 80 and 100 nm in diameter, and the pore depth of NIMS was 3.5 µm (**B**). The average pore size of DIOS was 120 nm, and the average pore depth was 450 nm. The image of the DIOS surface (**C,D**) was reprinted with permission from Ref. [23]. Copyright (2015), Elsevier.

### 3.3. NIMS-HR-MS for EVs Peptide Mass Fingerprint Identification

NIMS-MS allowed detection of potential peptide signals discriminating CFI and CRC group. Initially, peptide *m/z* spectral values acquired from the CRC, CLM CFI were statistically compared using the AUC interpretation method as shown in Supplementary Material Figure S1. SCiLS Lab software was used to identify spectral values that discriminate the disease groups, such as (a) CRC compared with CFI and (b) CLM compared with CFI. The linear discriminant analysis algorithm enabled comparative analysis between groups. Subsequently, the AUC method was used to evaluate the diagnostic ability of peptide ions in discriminating sample groups. AUC determined true positive and false positive ions and identified the top ions with AUC values ranging between 0.7 and 1. Subsequently,

by means of NIMS we detected 63 peptide ions (Table S1) able to discriminate the CRC and CFI group. For classifying CLM vs. CFI group, via NIMS we detected 56 peptide ions (Table S2). These peptide values were used for peptide sequence matching using MASCOT. MASCOT executes the MOSWE scoring algorithm to rank the top protein hits by comparing $5.6 \times 10^5$ peptide sequences from the SWISSPROT database and $2.03 \times 10^4$ sequences from the Homo sapiens (Uniprot ID—UP5640-H-sapiens) database. A panel of the top four putative proteins upregulated in plasma EVs is listed in Supplementary Materials Tables S3 and S4.

Peptide ions with the dominant ion observed at $m/z$ 3408.43 showed a match with the PP2A protein, determined to be a top protein upregulated in our CRC EVs samples (Figure 3A). Interestingly, evidence has accumulated that PP2A upregulation is strongly associated with the growth of CRC cells, as confirmed in tumour-adjacent tissues collected from patients [28–30]. When comparing CLM and CFI samples, peptide ions observed at $m/z$ 1770.98 and $m/z$ 3822.05 were 26% matched with ATF3 protein—determined to be upregulated in CLM EVs (Figure 3B).

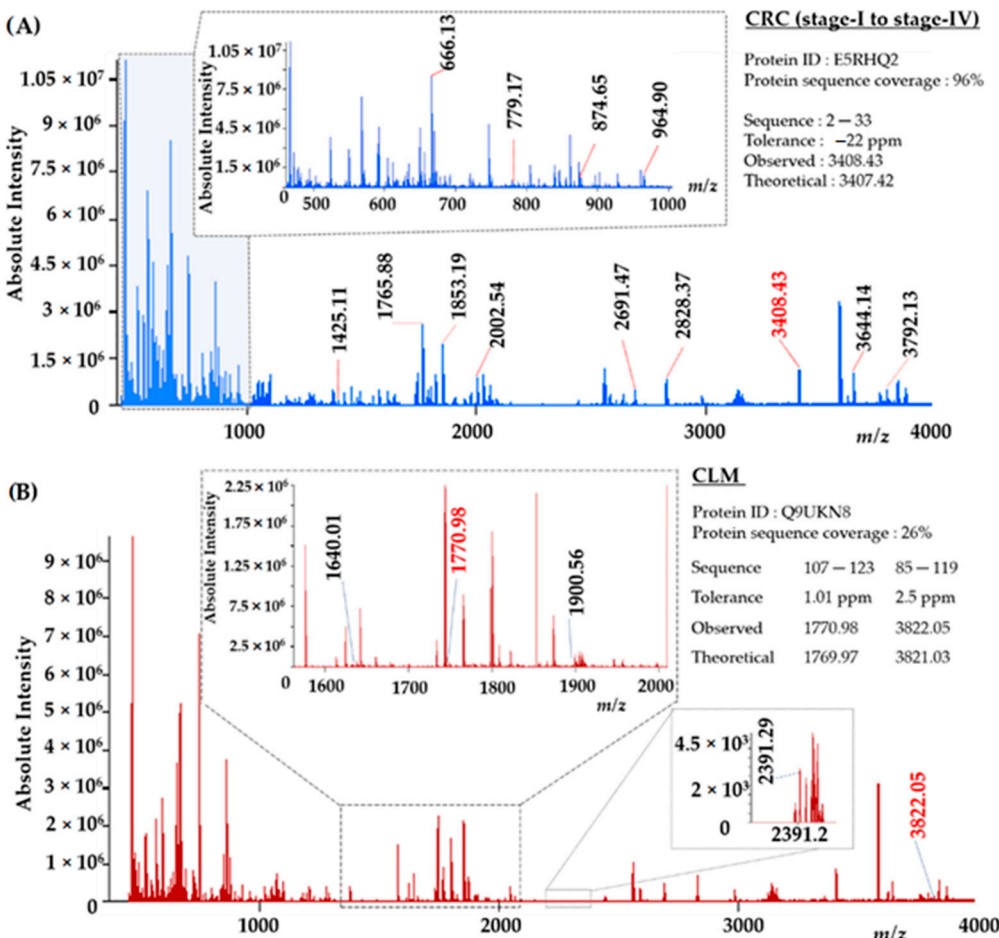

**Figure 3.** NIMS-FT-ICR-MS identified the top discriminative ions that can differentiate CRC disease from the CFI group. We interpreted the $m/z$ spectral value of peptide ions (labelled in red) that showed a high score of sequence coverage, which matches protein hits (MASCOT). (**A**) We identified $m/z$ 3408.43 ($\pm$2 ppm) matching PP2A was upregulated in CRC all stages (stage-I to IV) group compared to CFI. (**B**) Ions $m/z$ 1770.98 ($\pm$1.01 ppm) and $m/z$ 3822.05 ($\pm$2.05 ppm) matching ATF3 protein was upregulated in the CLM group compared to CFI.

### 3.4. Interpretation of Lipid Species Associated with Metastatic CRC Pathology Using DIOS-FT-ICR-MS

DIOS-MS allowed detection of more than 850 putative lithiated-lipid ions in each sample. As previously mentioned, linear discriminant analysis was performed using the in-built model algorithm in SCiLS Lab software to identify top discriminative ions. The diagnostic ability of spectral ions was assessed using a threshold AUC (>0.7) to compare ions that allow disease group classification. When compared to CFI group, we identified the top the level of 48 potential lithiated lipid ions intensity was increased in the CRC group, whereas, when compared to CFI the level of top 8 potential lithiated lipid ions intensity was increased in the CLM group. Subsequently, the ions above the AUC threshold were further validated in a publicly available database, LipidMaps. The *m/z* of precursor ion peak lists were compared against a variety of lipid classes. We observed that 23% of the phosphatidylcholine (PC) subset, 13% of the ceramide subset and 14% of the phosphatidylethanolamine subset were increased in CRC compared to the CFI group. When analysing CLM lipid ions, we observed that 27% of the phosphatidylinositol (PI) subset, 20% of the PG subset and 13% of the cholesterol-based (CL) subset were increased in the CLM compared to the CFI group. The list of identified putative lithiated lipids is shown in Supplementary Materials Tables S5 and S6.

Further, we performed a structural determination of some low abundant lithiated glycerophospholipids such as lysophosphatidylserine (LPS)-LPS (26:2) [M + Li]$^+$ observed at *m/z* 640.41, phosphatidylglycerol (PG)-PG (O-34:0; O) [M + Li]$^+$ at *m/z* 759.57 and PC (38:4) [M + Li]+ at *m/z* 816.60 in CRC compared to CFI. Figure 4A shows lithiated lipid peaks identified in CRC group. Further, direct analysis on DIOS can be used for structural characterisation of lipids through laser-induced dissociation (LID-MS/MS), where lithiated-PG (26:3) [M + Li]$^+$ has an sn1–acyl group with 18 carbon fatty acids containing 3 double bonds and an sn-2–acyl group with 8 carbon fatty acids anionic glycerol head group observed at *m/z* 639.37 [M + Li]$^+$ with mass tolerance 0.0074 Da. Figure 4B shows lithiated lipid peaks identified in the CLM group.

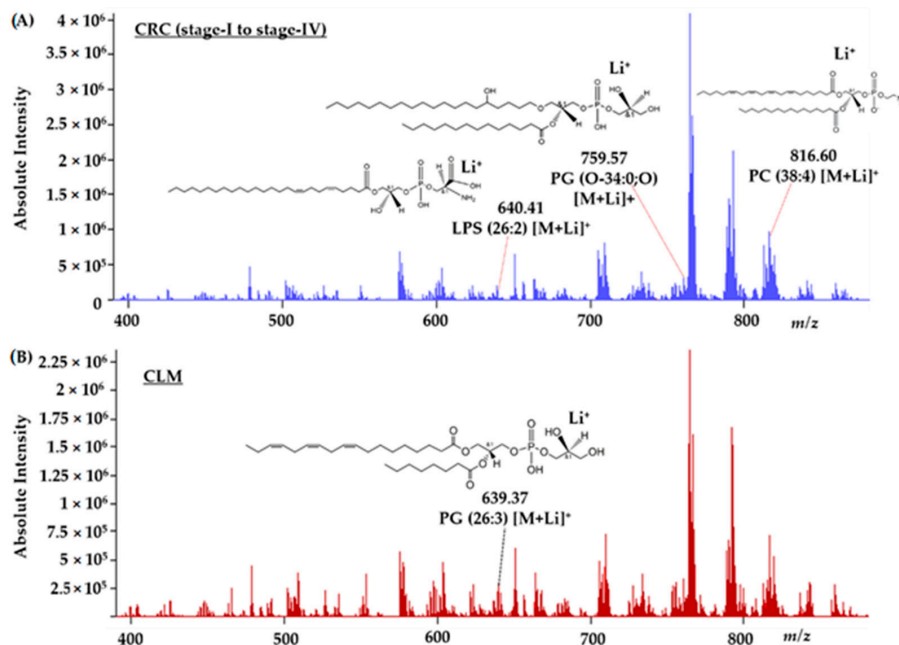

**Figure 4.** DIOS-HR-MS identified putative lithiated lipid subsets in plasma EVs. (**A**) When evaluating CRC group, the ions observed at *m/z* 640.41, *m/z* 759.57 and *m/z* 816.60 were matched with LPS (26:2) [M + Li]$^+$, PG (O-340;O) [M + Li]$^+$ and PC (38:4) [M + Li]$^+$, annotated in LipidMaps. These lipids were increased in intensity in the CRC group compared to CFI. (**B**) The peak observed at *m/z* 639.37, annotated as PG (26:3) [M + Li]$^+$, was increased in intensity in the CLM group compared to the CFI group.

## 4. Discussion

CRC cells release a larger number of EVs compared to healthy cells [31]. These EVs are enriched with tumour regulating signatures, circulating in human blood. Detection of such molecules in a low volume of patient samples is challenging. However, pSi SALDI surfaces such as NIMS and DIOS offer enhanced sensitivity to detect low abundant molecular signatures in blood EVs, useful for rapid and high throughput diagnosis of metastatic CRC progression.

NIMS-HR-MS identified that overexpression of PPA2 protein might be involved in CRC proliferation. The beta regulatory subunit of PP2A has been proposed to modulate catalytic activity and promote programmed cell death by initiating a series of signalling pathway events involved in the execution phase of cell apoptosis [32]. We observed that other EVs proteins, such as Toll-like receptors (TRL), may promote CRC progression in an inflammatory pathway. We have also found that the TLR1 protein is more upregulated in CRC patients, in 76% of the plasma samples, compared to healthy individuals – suggesting inflammatory pathways are activated in CRC [33]. Furthermore, we observed that the ATF3 protein is upregulated in CLM EVs. Several studies have proposed that ATF3 functions as a tumour suppressor in CRC and one of the significantly over-represented transcription factors involved in CRC progression [34,35]. A study investigated by Hackl et al. reported that ATF3 expression is upregulated in human CRC tissue compared to healthy individuals [34]. However, the actual mechanism of how ATF3 promotes metastasis is yet to be confirmed. Interestingly, NIMS-HR-MS identified ATF3 protein in the plasma EVs of CLM samples but did not identify significant changes (above the AUC threshold) in the CRC group.

The addition of lithium to the EVs samples reduced mass spectral complexity for lipid identification via DIOS-HR-MS. The results showed an evident shift in the profile of glycerophospholipid and sphingolipids in CRC and CLM groups, who had been diagnosed with dysplasia in the colon. A study conducted by Shen et al. identified several phospholipids, including PG (34:0), lysoPC (18:2), PE (O-36:3), and sphingolipids such as SM (38:8) and Cer (44:5) in plasma EVs as a potential biomarkers for CRC diagnosis [36]. Our observation agrees with this previous study, and we have detected that a wide range of putative lithiated-phospholipids including PG (O-34:0), PC (36:4), PE (O-38:5) and lithiated sphingolipids SM (36:1) and Cer (34:2) were increased in intensity in CRC group plasma EVs. According to the ceramide–sphingosine–S1P rheostat model, both SM and Cer promote apoptosis [37,38]; our results also concur that altered sphingolipid metabolism may contribute to cancer progression. Interestingly, when comparing CLM with CFI group, a significant increase in intensity of CL (68:14) in a CLM group depicted that CL subsets may enhance metastatic CRC progression, possibly via reactive oxidative species and the MAPK signalling pathway [39]. In addition, we could speculate that PI subclasses, upregulated in CLM, modulate phosphatidylinositol-3 kinase (PI3K) enzyme function, which permits the phosphorylation pathway in metastatic progression. Several studies reported that the PI3K pathway is triggered in metastatic CRC progression [40,41]. However, further analyses need to be performed in a larger number of plasma samples to confirm the identified lipids were significantly increased in CRC compared to CFI; and a cross verification of any enzyme activity related to MAPK and PI3K pathway, is still required.

Furthermore, traditional chromatographic MS approaches are difficult to perform at scale in a low volume of biological samples. A panel of putative lipids with a mass tolerance of ±0.005 Da shows that DIOS-HR-MS is useful for profiling low volume samples at speed, enabling hundreds of samples to be profiled in minutes. Although the precise mechanisms of lipids in CRC have not been elucidated, the generation of lithiated ions has allowed us to putatively identify lipids involved in CRC progression, which could be applied to large biobanked sample cohorts.

## 5. Conclusions

We have profiled a range of potential CRC biomarkers in <100 µL biobank stored blood samples. Nanostructured pSi SALDI-HR-MS enabled a snapshot of the molecular landscape of CRC progression using a low volume of the patient sample—allowing a chemical description of CRC heterogeneity. Using SALDI-HR-MS we identified putative lipids including PG (O-34:0), PC (36:4), PE (O-38:5), SM (36:1) and Cer (34:2) and proteins such as PP2A and ATF3 that can discriminate disease groups and could act as potential biomarkers for CRC and CLM classification and diagnosis. Moreover, SALDI-HR-MS offers rapid detection of lipids in biological samples without performing time-consuming chromatographic separation and labelling techniques. Based on our results, we expect that SALDI-HR-MS can be expanded to other heterogeneous disease subtype classifications after validation of the corresponding biomarkers.

**Supplementary Materials:** The following supporting information can be downloaded at: https://www.mdpi.com/article/10.3390/jnt3040013/s1, Figure S1: Workflow of small extracellular vesicles (EVs) biomarker detection using SALDI-MS for CRC diagnosis. (A) Plasma EVs were isolated from metastatic CRC and CFI individuals. Then trypsin-digestion proteins and lithium adduct lipid samples were spotted on NIMS, and DIOS surfaces, respectively. (B) SALDI surfaces functionalised with a fluorinated silane are then coupled with a high-resolution mass spectrometer. CRC—colorectal cancer stage-I to stage-IV; CFI—cancer free individual and healthy participants; pSi—Porous silicon; NIMS—Nanostructure-initiator mass spectrometry; SALDI—Surface-assisted laser desorption/ionisation mass spectrometry; DIOS—Desorption/ionisation on porous silicon; Table S1: Top 63 peptide ions discriminate CRC and CFI group. CRC—colorectal cancer stage-I to stage-IV; CFI—cancer free individuals and healthy participants; Table S2: Top 56 peptide ions discriminate CLM and CFI group. CLM—colorectal cancer liver metastasis; CFI—cancer free individuals and healthy participants; Table S3: Top 4 discriminative ions identified by NIMS-FT-ICR-MS were applied in the Mascot peptide mass fingerprint for protein matching. We identified the top-five potential protein signatures that can differentiate CRC and CFI group. CRC—colorectal cancer stage-I to stage-IV; CFI—cancer free individuals and healthy participants; NIMS-FT-ICR-MS—Nanostructure-initiator mass spectrometry—Fourier-transform ion cyclotron resonance mass spectrometry; Table S4: Top 4 discriminative ions identified by NIMS-FT-ICR-MS were applied in the Mascot peptide mass fingerprint for the protein match. We identified top-five potential protein signatures that can differentiate CLM and CFI group. CLM—colorectal cancer liver metastasis; CFI—cancer free individuals and healthy participants; NIMS-FT-ICR-MS—Nanostructure-initiator mass spectrometry—Fourier-transform ion cyclotron resonance mass spectrometry; Table S5: Top discriminative ions were identified by DIOS-FT-ICR-MS, which was applied in the LipidMaps database for annotation. We identified a profile of lithiated-lipids that can differentiate CRC from the CFI group. CRC—colorectal cancer stage-I to stage-IV; CFI—cancer free individuals and healthy participants; DIOS-FT-ICR-MS—desorption/ionisation on porous silicon fourier-transform ion cyclotron resonance mass spectrometry. LPE—lysophosphatidylethanolamine, Cer—Ceramide, NAE—N-acylethanolamines, CerP—Ceramide 1-phosphates, LPS—lysophosphatidylserine, PC—phosphatidylcholine, PS—phosphatidylserine, LPC—lysophosphatidylcholine, SM—sphingomyelin, PE—phosphatidylethanolamine, DG—diacylglycerol, PG—phosphatidylglycerol, LPG—lysophosphatidylglycerol, LPI—lysphosphatidylionisitol, TG—triacylglycerol; Table S6: DIOS-FT-ICR-MS identified top discriminative ions, annotated using LipidMaps. We identified a profile of lithiated-lipids that can differentiate CLM and CFI group. CLM—colorectal cancer liver metastasis; CFI—cancer free individual and healthy participants; DIOS-FT-ICR-MS—desorption/ionisation on porous silicon fourier-transform ion cyclotron resonance mass spectrometry; PS—phosphatidylserine, PG—phosphatidylglycerol, LPG—lysophosphatidylglycerol, PI—phosphatidylinositol, TG—triacylglycerol, CL—cholesterol, PA—phosphatidicacid

**Author Contributions:** Conceptualization, S.T.K., D.R., E.H. and N.H.V.; methodology, S.T.K., D.R., R.R., E.E.A. and R.S.M.; software, S.T.K. and D.R.; validation, S.T.K., D.R. and E.E.A.; formal analysis, S.T.K., D.R., R.R. and R.S.M.; investigation, S.T.K., D.R., E.H. and N.H.V.; resources, C.K., G.J.M., K.F. and E.H.; data curation, S.T.K. and D.R.; writing—original draft preparation, S.T.K. and D.R.; writing— S.T.K., D.R. and N.H.V.; visualization, S.T.K. and D.R.; supervision, D.R. and N.H.V.; project administration, D.R., C.K., G.J.M., K.F., E.H. and N.H.V.; funding acquisition, N.H.V. All authors have read and agreed to the published version of the manuscript.

**Funding:** S.T.K. was supported by a Monash Graduate Scholarship, Monash University, Australia.

**Institutional Review Board Statement:** The study was approved by the Ethics Committee of Monash University (Protocol code MUHREC 17194; 2019) and The University of Adelaide (Protocol code HREC/14/TQEHLMH/164).

**Informed Consent Statement:** Informed consent was obtained by The University of Adelaide from all subjects involved in the study.

**Data Availability Statement:** MS files are made available as Supplementary Materials.

**Acknowledgments:** This work was performed in part at the Melbourne Centre for Nanofabrication (MCN) in the Victorian Node of the Australian National Fabrication Facility (ANFF). The authors would like to acknowledge Jacinta White from the CSIRO Manufacturing Morphology and Structure Group within the Materials Characterisation and Modelling Program for TEM contribution to this project. E.E.A. acknowledges the fellowship granted by CONACYT (CVU 336883, Becas de Consolidacion modalidad Repatriacion 2021).

**Conflicts of Interest:** The authors declare no conflict of interest.

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
