# Peer review of "Nanostructured Silicon Enabled HR-MS for the Label-Free Detection of Biomarkers in Colorectal Cancer Plasma Small Extracellular Vesicles"

_jnt, doi:10.3390/jnt3040013_

Round 1

Reviewer 1 Report

The manuscript Nanostructured silicon enabled HR-MS for the label-free detection of biomarkers in colorectal cancer plasma exosomes

Please include data about exosomes characterization in the main figure?

Author Response

As suggested, exosome characterisation data including TEM and NTA datasets have been added in the Figure 1, Panel 1 (row 290) and Figure 1, Panel 2 (row 296).

Reviewer 2 Report

The project is interesting and will be useful for the community researching vesicles and considering their application in medicine. 

1)     Since the definition of exosomes is evolving it might more correct to name the vesicles you were working with as small extracellular vesicles (EVs) since exosomes are considered to only be a subset of small EVs.

2)     Is there a reason of why you chose Total Isolation Reagent (Invitrogen 4478359) that is more for cell culture media and not Total Isolation Reagent 4484450 that is designed for isolation from plasma?

3)     Did you mean NanoSight NS300 instead of NanoSight N300?

4)     Can you please give a bit more detail about NTA parameters (intensity settings, threshold, etc)?

5)     Please double check if 1500 g is ok for this journal. Maybe use 1500xg for centrifugation description in the methods section?

6)     At row 188 remove dead space here “thiols .” Please double check spelling and spaces throughout.

7)     You report concentration determined by NTA as 3.27e+08 but my understanding this was the concentration after you made NTA measurements. Can you take into account the dilution used for analysis in order to report the exosome concentration you had after isolation? This may be more useful for the reader.

8)     The difference is size obtained by NTA and TEM was previously discussed here: https://pubmed.ncbi.nlm.nih.gov/25821114/ and is more based on how measurements are done. With NTA you use light scattering and obtain the hydrodynamic size while with TEM you obtain the geometric size.

9)     Have you adjusted the exosome concentration after isolation and before the biomarker detection? To make a side by side comparison it would have been useful to keep at least the total exosome concentration or some other parameter in samples the same (e.g. based on NTA analysis or total protein concentration). If you have analyzed NTA of all exosome samples from plasma it would be useful to present it in the SI. Would be useful to know what the exosome concentration was in different patients and controls. 

10)  It is of common practice to evaluate how successful the exosome isolation was. Would be good to show a western blot/flow cytometry data of a few typical exosome membrane markers (e.g. cd63, CD9 and or CD81) to make sure your isolation was successful.

It may be useful for you to go over this paper https://pubmed.ncbi.nlm.nih.gov/30637094/. Especially note the criteria when working with plasma EVs. 

Author Response

The project is interesting and will be useful for the community researching vesicles and considering their application in medicine. 

1)     Since the definition of exosomes is evolving it might more correct to name the vesicles you were working with as small extracellular vesicles (EVs) since exosomes are considered to only be a subset of small EVs.

As suggested, exosomes were renamed as small extracellular vesicles (EVs) within the main text and through-out the manuscript

2)     Is there a reason of why you chose Total Isolation Reagent (Invitrogen 4478359) that is more for cell culture media and not Total Isolation Reagent 4484450 that is designed for isolation from plasma?

Our preliminary tests to determine the number of EVs required for effective nano-Si HR-MS were achieved using cell cultures, as the small amount of available patient plasma is considered highly valuable and not suitable for optimisation purposes. As we developed the parameters using cell cultures, we then continued using the same reagent that had initially proved successful.

3)     Did you mean NanoSight NS300 instead of NanoSight N300?

The instrument name has been changed to NanoSight NS300.

4)     Can you please give a bit more detail about NTA parameters (intensity settings, threshold, etc)?

To maintain comparable results, the same settings were used throughout all experiments: detection threshold was set at 2, camera level 8, and a media dilution of 1:1000. Detail about NTA settings were added in the section 2.3, row 161 to 163.

5)     Please double check if 1500 g is ok for this journal. Maybe use 1500xg for centrifugation description in the methods section?

The centrifugation force speed has been updated to 1500 x g in row 181. Also the relative centrifugal force have been updated throughout the documents.

6)     At row 188 remove dead space here “thiols .” Please double check spelling and spaces throughout.

At row 189, the space in “thiols” has been removed.

7)     You report concentration determined by NTA as 3.27e+08 but my understanding this was the concentration after you made NTA measurements. Can you take into account the dilution used for analysis in order to report the exosome concentration you had after isolation? This may be more useful for the reader.

The dilution used for analysis after isolation was 1:1000 ─ mentioned in row 162.

8)     The difference is size obtained by NTA and TEM was previously discussed here: https://pubmed.ncbi.nlm.nih.gov/25821114/ and is more based on how measurements are done. With NTA you use light scattering and obtain the hydrodynamic size while with TEM you obtain the geometric size.

As suggested, for NTA hydrodynamic size of Mean: 152.0 +/- 1.1 nm, Mode: 125.7 +/- 5.8 nm, SD: 43.1 +/- 2.1 nm were added to Figure 1 row number 298 and 299, and for TEM plasma-derived EVs ranged between 40 to 120 nm size with spheroid shape were added in the row number 299 and 300.

9)     Have you adjusted the exosome concentration after isolation and before the biomarker detection? To make a side by side comparison it would have been useful to keep at least the total exosome concentration or some other parameter in samples the same (e.g. based on NTA analysis or total protein concentration). If you have analyzed NTA of all exosome samples from plasma it would be useful to present it in the SI. Would be useful to know what the exosome concentration was in different patients and controls. 

We have used BCA assay to measure the total protein concentration in each sample (both metastatic CRC and healthy) to ensure variation from MS detection thresholds don’t confound disease vs control differences. Subsequently, only sample aliquots containing a minimum of 400 ug of proteins were used for MS analysis. We agree, if we were to have more plasma available a comprehensive comparison of total EV content in each group would be additionally informative, particularly in the pre-metastasis stage of progression.  

10)  It is of common practice to evaluate how successful the exosome isolation was. Would be good to show a western blot/flow cytometry data of a few typical exosome membrane markers (e.g. cd63, CD9 and or CD81) to make sure your isolation was successful.

It may be useful for you to go over this paper https://pubmed.ncbi.nlm.nih.gov/30637094/. Especially note the criteria when working with plasma EVs. 

This study used an established exosome isolation procedure (Invitrogen 4478359) and we relied on characterisation using TEM and NTA to determine successful isolation of exosomes from blood-plasma samples, as these tools are readily available to us.  Our approach has been used previously, where a study reported by RongXu et al 2015, Methods (https://doi.org/10.1016/j.ymeth.2015.04.008), used ultra-centrifugation to isolate and identified that T-complex protein 1 subunit was enriched in human colon cancer cell lines. Our study also confirmed that T-complex protein 1 subunit was upregulated in patient metastatic colorectal cancer.

Reviewer 3 Report

In this work, the authers reported pre-characterised biobanked plasma samples from surgical units, typically with a low volume (~100 µL), to generate and discover signatures of CRC-derived exosomes. They employed nanostructured porous silicon (pSi) surface assisted-laser desorption/ionisation (SALDI) coupled with high-resolution mass spectrometry (HR-MS), to allow sensitive detection of low abundant analytes in plasma exosomes. They observed that lithium chloride enhanced detection sensitivity to elucidate the structure of low abundant lipids in plasma exosomes. pSi SALDI can be used as an effective system for label-free and high throughput analysis of low-volume patient samples, allowing rapid and sensitive analysis for CRC classification. This work is interesting but needs a bit revision before it will be suitable for publication.

1. In line 185, the pellets were incubated at 95°C for 10 min to denature the remaining intact 185

proteins. The targeted protein also were denatured. This may influence the result.

2. It is suggested to compare with the previous studies.

Author Response

  1. In line 185, the pellets were incubated at 95°C for 10 min to denature the remaining intact 185

proteins. The targeted protein also were denatured. This may influence the result.

It is necessary to denature proteins prior to effective trypsin digestion; particularly where low quantities are involved. Additionally, we have added reducing and alkylating agents to break disulfide bridges, followed by trypsin enzyme digestion to cleave arginine and lysine sites, allowing identification of peptide sequences that related to specific proteins in the Mascot database. Without denaturing, reduction and alkylation, peptides containing disulfide bonds and complex quaternary structures would be difficult to identify during database searching.

  1. It is suggested to compare with the previous studies.

We have used well established proteomic protocols developed within the Monash Proteomics & Metabolomics Facility. These same procedures have been used to publish in leading journals (see https://www.monash.edu/researchinfrastructure/mpmf/downloads)
